# DiscEval: Discourse Based Evaluation of Natural Language Understanding

## Abstract

New models for natural language understanding have made unusual progress recently, leading to claims of universal text representations. However, current benchmarks are predominantly targeting semantic phenomena; we make the case that discourse and pragmatics need to take center stage in the evaluation of natural language understanding. We introduce DiscEval, a new benchmark for the evaluation of natural language understanding, that unites 11 discourse-focused evaluation datasets. DiscEval can be used as supplementary training data in a multi-task learning setup, and is publicly available, alongside the code for gathering and preprocessing the datasets. Using our evaluation suite, we show that natural language inference, a widely used pretraining task, does not result in genuinely universal representations, which opens a new challenge for multi-task learning.

## 1 Introduction

Over the last year, novel models for natural language understanding (NLU) have made a remarkable amount of progress on a number of widely accepted evaluation benchmarks. The GLUE benchmark (Wang et al., 2018), for example, was designed to be a set of challenging NLU tasks, such as question answering, sentiment analysis, and textual entailment; yet, current state of the art systems surpass human performance estimates on the average score of its subtasks (Yang et al., 2019). Similarly, the NLU subtasks that are part of the SentEval framework, a widely used benchmark for the evaluation of sentence-to-vector encoders, are successfully dealt with by current neural models, with scores that exceed the 90% mark.[1]

The results on these benchmarks are impressive, but sometimes lead to excessive optimism regarding the ability of current NLU models. For example, based on the resulting performance on the above-mentioned benchmarks, a considerable number of researchers has even put forward the claim that their models induce *universal* representations (Cer et al., 2018; Kiros & Chan, 2018; Subramanian et al., 2018; Wieting et al., 2015; Liu et al., 2019). It is important to note, however, that benchmarks like SentEval and GLUE are primarily focusing on semantic aspects, i.e. the literal and uncontextualized content of text. While the semantics of language is without doubt an important aspect of language, we believe that a single focus on semantic aspects leads to an impoverished model of language.

For a versatile model of language, other aspects of language, viz. pragmatic aspects, equally need to be taken into account. Pragmatics focuses on the larger context that surrounds a particular textual instance, and they are central to meaning representations that aspire to lay a claim to universality. Consider the following utterance :

(1)    You're standing on my foot.

The utterance in (1) has a number of direct implications that are logically entailed by the utterance above, such as the implication that the hearer is standing on a body part of the speaker, and the implication that the speaker is touching the hearer. But there are also more indirect implications, that are not literally expressed, but need to be inferred from the context, such as the implication that the speaker wants the hearer to move away from them. The latter kind of implication, that is indirectly

---

[1]http://nlpprogress.com/english/semantic_textual_similarity.html

implied by the context of an utterance, is called *implicature*—a term coined by Grice (1975). In real world applications, recognizing the implicatures of a statement is arguably more important than recognizing its mere semantic content.

The implicatures that are conveyed by an utterance are highly dependent on its illocutionary force (Austin, 1975). In Austin's framework, the *locution* is the literal meaning of an utterance, while the *illocution* is the goal that the utterance tries to achieve. When we restrict the meaning of (1) to its locution, the utterance is reduced to the mere statement that the hearer is standing on the speaker's foot. However, when we also take its illocution into account, it becomes clear that the speaker actually formulates the request that the speaker step away. The utterance's illocution is clearly an important part of the entire meaning of the utterance, that is complementary to the literal content (Green, 2000).[2]

The example above makes clear that pragmatics is a fundamental aspect of the meaning of an utterance. Semantics focuses on the literal content of utterances, but not on the kind of goal the speaker is trying to achieve. Pragmatic (i.e. discourse-based) tasks focus on the actual use of language, so a discourse-centric evaluation could *by construction* be a better fit to evaluate how NLU models perform in practical use cases, or at least should be used as a complement to semantics-focused evaluations benchmarks. Ultimately, many use cases of NLP models are related to conversation with end users or analysis of structured documents. In such cases, discourse analysis (i.e. the ability to parse high-level textual structures that take into account the global context) is a prerequisite for human level performance. Moreover, standard benchmarks often strongly influence the evolution of NLU models, which means they should be as exhaustive as possible, and closely related to the models' end use cases.

In this work, we compile a list of 11 discourse-focused tasks that are meant to complement existing benchmarks. We propose: (i) A new evaluation benchmark, named *DiscEval*, which we make publicly available.[3] (ii) Derivations of human accuracy estimates for some of the tasks. (iii) Evaluation on these tasks of a state of the art generalizable NLU model, viz. BERT, alongside BERT augmented with auxiliary finetunings. (iv) New comparisons of discourse-based and Natural Language Inference based training signals showing that the most widely used auxiliary finetuning dataset, viz. MNLI, is not the best performing on DiscEval, which suggests a margin for improvements.

## 2 RELATED WORK

Evaluation methods of NLU have been the object of heated debates since the proposal of the Turing Test. Automatic evaluations relying on annotated datasets are arguably limited but they became a standard. They can be based on sentence similarity (Agirre et al., 2012), leveraging human annotated scores of similarity between sentence pairs. Predicting similarity between two sentences requires some representation of their semantic content beyond their surface form, and sentence similarity estimation tasks can potentially encompass many aspects, but it is not clear how humans annotators weight semantic, stylistic, and discursive aspects while rating.

Using a set of more focused and clearly defined tasks has been a popular approach. Kiros et al. (2015) proposed a set of tasks and tools for sentence understanding evaluation. These 13 tasks were compiled in the SentEval (Conneau et al., 2017) evaluation suite designed for automatic evaluation of pre-trained sentence embeddings. SentEval tasks are mostly based on sentiment analysis, semantic sentence similarity and natural language inference. Since SentEval evaluates sentence embeddings, the users have to provide a sentence encoder that is not finetuned during the evaluation.

GLUE (Wang et al., 2018) proposes to evaluate language understanding with less constraints than SentEval, allowing users not to rely on explicit sentence embedding based models. They compile 9 classification or regression tasks that are carried out for sentences or sentence pairs. 3 tasks focus on semantic similarity, and 4 tasks are based on NLI, which makes GLUE arguably semantics-based,

---

[2]In order to precisely determine their illocution, utterances have been categorized into classes called speech acts (Searle et al., 1980), such as ASSERTION, QUESTION or ORDER which have different kinds of effects on the world. For instance, constative speech acts (e.g. *the sky is blue*) describe a state of the world and are either true or false while performative speech acts (e.g. *I declare you husband and wife*) can change the world upon utterance (Austin, 1975).

[3]https://github.com/disceval/DiscEval

even though it also includes sentiment (Socher et al., 2013) and grammaticality (Warstadt et al., 2018).

NLI can be regarded as a universal framework for evaluation. In the *Recast* framework (Poliak et al., 2018), existing datasets (e.g. sentiment analysis) are formulated as NLI tasks. For instance, based on the sentence *don't waste your money*, annotated as a negative review, they use handcrafted rules to generate the following example: (PREMISE: *When asked about the product, liam said "don't waste your money"* , HYPOTHESIS: *Liam didn't like the product*, LABEL: entailment). However, the generated datasets prevent the evaluation to measure directly how well a model deals with the semantic phenomena present in the original dataset, since some sentences use artificially generated reported speech. Thus, NLI data could be used to evaluate discourse analysis, but it is not clear how to generate examples that are not overly artificial. Moreover, it is unclear to what extent instances in existing NLI datasets need to deal with pragmatic aspects (Bowman, 2016).

SuperGLUE (Wang et al., 2018) updates GLUE with six novel tasks that are selected to be even more challenging. Two of them deal with contextualized lexical semantics, two tasks are a form of question answering, and two of them are NLI problems. One of those NLI tasks, CommitmentBank (de Marneffe et al., 2019), is the only explicitly discourse-related task.

Another effort towards evaluation of general purpose NLP systems is DecaNLP (McCann et al., 2018). The 10 tasks of this benchmarks are all framed as question answering. For example, a question answering task is derived from a sentiment analysis task using artificial questions such as *Is this sentence positive or negative?* Four of these tasks deal with semantic parsing, and other tasks include NLI and sentiment analysis. Discourse phenomena can be involved in some tasks (e.g. the summarization task) although it is hard to assess to what extent.

Discourse relation prediction has punctually been used for sentence representation learning evaluation, by Nie et al. (2019) and Sileo et al. (2019), but they all used only one dataset (viz. the PDTB (Prasad et al., 2008)), which we included in our benchmark. Discourse has also been considered for evaluation in the field of machine translation. Läubli et al. (2018) showed that neural models achieve superhuman results on sentence-level translations but that current models yield underwhelming results when considering document-level translations, also making a case for discourse-aware evaluations.

Other evaluations, such as linguistic probing or GLUE diagnostics (Conneau et al., 2018; Belinkov & Glass, 2019; Wang et al., 2019b), focus on an internal understanding of what is captured by the models (e.g. syntax, lexical content), rather than measuring performance on external tasks; they provide a complementary viewpoint, but are outside the scope of this work.

## 3 PROPOSED TASKS

Our goal is to compile a set of diverse discourse-related tasks. We restrict ourselves to classification either of sentences or sentence pairs and only use publicly available datasets that are absent from other benchmarks (SentEval/GLUE/SuperGLUE).

The scores in our tasks are not all meant to be compared to previous work, since we alter some datasets to yield more meaningful evaluations (we perform duplicate removal or class subsampling when mentioned). We found these operations necessary in order to leverage the rare classes and yield more meaningful scores. As an illustration, GUM initially consists of more than $99\%$ of *unattached* labels, and SwitchBoard contains $80\%$ of *statements*.

We first present the tasks we selected, also described in table 1 and appendix A, and then propose a rudimentary taxonomy of how they address different aspects of meaning.

**PDTB** The Penn Discourse Tree Bank (Prasad et al., 2014) contains a collection of fine-grained implicit (i.e. not signaled by a discourse marker) relations between sentences from the news domain in the Penn TreeBank 2.0. We select the level 2 (called types in PDTB terminology) relations as categories.

**STAC** (Strategic Conversation; Asher et al., 2016) is a corpus of strategic chat conversations manually annotated with negotiation-related information, dialogue acts and discourse structures in the framework of Segmented Discourse Representation Theory (SDRT; Asher & Lascarides, 2003). We only consider pairwise relations between all dialog acts, following Badene et al. (2019). We

| dataset | categories | example | class | $N_{train}$ |
|---|---|---|---|---|
| PDTB | discourse relation | "it was censorship"/"it was outrageous" | conjunction | 13k |
| STAC | discourse relation | "what ?"/"i literally lost" | question-answer-pair | 11k |
| GUM | discourse relation | "Do not drink"/"if underage in your country" | condition | 2k |
| Emergent | stance | "a meteorite landed in nicaragua."/"small meteorite hits managua" | for | 2k |
| SwitchBoard | speech act | "well , a little different , actually ," | hedge | 19k |
| MRDA | speech act | "yeah that 's that 's that 's what i meant ." | acknowledge-answer | 14k |
| Persuasion | E/S/S/R | "Co-operation is essential for team work"/"lions hunt in a team" | low specificity | 0.6k |
| SarcasmV2 | sarcasm presence | "don't quit your day job"/"[...] i was going to sell this joke. [...]" | sarcasm | 9k |
| Squinky | I/I/F | "boo ya." | uninformative, high implicature, unformal | 4k |
| Verifiability | verifiability | "I've been a physician for 20 years." | verifiable-experiential | 6k |
| EmoBank | V/A/D | "I wanted to be there.." | low valence, high arousal, low dominance | 5k |

Table 1: DiscEval classification datasets. $N_{train}$ is the number of examples in the training set. E/S/S/R denotes Eloquence/Strength/Specificity/Relevance; I/I/F is Information/Implicature/Formality; V/A/D denotes Valence/Arousal/Dominance

remove duplicate pairs and dialogues that only have non-linguistic utterances (coming from the game server). We subsample dialog act pairs with no relation so that they constitute 20% of each fold.

**GUM** (Zeldes, 2017) is a corpus of multilayer annotations for texts from various domains; it includes discourse structure annotations according to Rhetorical Structure Theory (RST; Mann & Thompson, 1987). Once again, we only consider pairwise interactions between discourse units (e.g. sentences/clauses). We subsample discourse units with no relation so that they constitute 20% of each document. We split the examples in train/test/dev sets randomly according to the document they belong to.

**Emergent** (Ferreira & Vlachos, 2016) is composed of pairs of assertions and titles of news articles that are *against*, *for*, or *neutral* with respect to the opinion of the assertion.

**SwitchBoard** (Godfrey et al., 1992) contains textual transcriptions of dialogs about various topics with annotated speech acts. We remove duplicate examples and subsample *Statements* and *Non Statements* so that they constitute 20% of the examples. We use a custom train/validation split (90/10 ratio) since our preprocessing leads to a drastic size reduction of the original development set. The label of a speech act can be dependent on the context (previous utterances), but we discarded it in this work for the sake of simplicity, even though integration of context could improve the scores (Ribeiro et al., 2015).

**MRDA** (Shriberg et al., 2004) contains textual transcription of multi-party real meetings, with speech act annotations. We remove duplicate examples. We use a custom train/validation split (90/10 ratio) since this deduplication leads to a drastic size reduction of the original development set, and we subsample *Statement* examples so that they constitute 20% of the dataset. We also discarded the context.

**Persuasion** (Carlile et al., 2018) is a collection of arguments from student essays annotated with factors of persuasiveness with respect to a claim; considered factors are the following: Specificity, Eloquence, Relevance and Strength. For each graded target, we cast the ratings into three quantiles and discard the middle quantile.

**SarcasmV2** (Oraby et al., 2016) consists of messages from online forums with responses that may or may not be sarcastic according to human annotations.

**Squinky dataset** (Lahiri, 2015) gathers annotations on Formality, Informativeness, and Implicature, where sentences were graded on a scale from 1 to 7. The Implicature score is defined as the amount of information that is not explicitly expressed in a sentence. For each target, we cast the ratings into three quantiles and discard the middle quantile.

**Verifiability** (Park & Cardie, 2014) is a collection of online user comments annotated as *Verifiable-Experiential* (verifiable and about writer's experience) *Verifiable-Non-Experiential* or *Unverifiable*.

**EmoBank** (Buechel & Hahn, 2017) aggregates emotion annotations on texts from various domains using the VAD representation format. The authors define Valence as *corresponding to the concept of polarity*[4], Arousal as *degree of calmness or excitement* and Dominance as *perceived degree of control over a situation*. For each target, we cast the ratings into three quantiles and discard the middle quantile.

It has been argued by Halliday (1985) that linguistic phenomena fall into three metafunctions: *ideational* for semantics, *interpersonal* for appeals to the hearer/reader, and *textual* for form-related aspects. This forms the basis of discourse relation types by Hovy & Maier (1992), who call them semantic, interpersonal and presentational. DiscEval tasks cut across these categories, because some of the tasks integrate all aspects when they characterize the speech act or discourse relation category associated to a discourse unit (mostly sentences), an utterance or a pair of these. However, most discourse relations involved focus on *ideational* aspects, which are thus complemented by tasks insisting on more interpersonal aspects (e.g. using appeal to emotions, or verifiable arguments) that help realize speech act intentions. Finally, intentions can achieve their goals with varying degrees of success. This leads us to a rudimentary grouping of our tasks:

– The speech act classification tasks (SwitchBoard, MRDA) deal with the detection of the intention of utterances. They use the same label set (viz. DASML, Allen & Core, 1997) but different domains and annotation guidelines. A discourse relation also characterizes how an utterance contributes to the coherence of a document/conversation (e.g through *elaboration* or *contrast*), so this task requires a form of understanding of the use of a sentence, and how a sentence fits with another sentence in a broader discourse. A discourse relation can be seen as a speech act whose definition is tied to a structured context (Asher & Lascarides, 2003). Here, three tasks (PDTB, STAC, GUM) deal with discourse relation prediction with varying domains and formalisms.[5] The Stance detection task can be seen as a coarse-grained discourse relation classification.

– Detecting emotional content, verifiability, formality, informativeness or sarcasm is necessary in order to figure out in what realm communication is occurring. A statement can be persuasive, yet poorly informative and unverifiable. Emotions (Dolan, 2002) and power perception (Pfeffer, 1981) can have a strong influence on human behavior and text interpretation. Manipulating emotions can be the main purpose of a speech act as well. Sarcasm is another means of communication and sarcasm detection is in itself a straightforward task for the evaluation of pragmatics, since sarcasm is a clear case of literal meaning being different from the intended meaning.

– Persuasiveness prediction is a useful tool to assess whether a model can measure how well a sentence can achieve its intended goal. This aspect is orthogonal to the determination of the goal itself, and is arguably equally important.

## 4 EVALUATIONS

### 4.1 MODELS

Our goal is to assess the performance of popular NLU models and the influence of various training signals on DiscEval scores. We evaluate state of the art models and baselines on DiscEval using the Jiant (Wang et al., 2019c) framework. Our baselines include average of GloVe (Pennington et al., 2014) embeddings (CBoW) and BiLSTM with GloVe and ELMo (Peters et al., 2018) embeddings. We also evaluate BERT (Devlin et al., 2019) base uncased models, and perform experiments with *Supplementary Training on Intermediate Labeled-data Tasks* (STILT; Phang et al., 2018). STILT is a further pretraining step on a data-rich task before the final fine-tuning evaluation on the target task. STILTs can be combined using multitask learning. We use Jiant default parameters[6], and uniform loss weighting when multitasking (a different task is optimized at each training batch).

We finetune BERT with four of such training signals:

---

[4]This is the dimension that is widely used in sentiment analysis.

[5]These formalisms have different assumptions about the nature of discourse structure.

[6]`https://github.com/nyu-mll/jiant/blob/706b6521c328cc3dd6d713cce2587ea2ff887a17/jiant/config/examples/stilts_example.conf`

**MNLI** (Williams et al., 2018) is a collection of 433k sentence pairs manually annotated with *contradiction*, *entailment*, or *neutral* relations. Phang et al. (2018) showed that finetuning with this dataset leads to accuracy improvement on all GLUE tasks except CoLA (Warstadt et al., 2018).

**DisSent** (Nie et al., 2019) consists of $4.7M$ sentence pairs that are separated by a discourse marker (from a list of 15 markers). Prediction of discourse markers based on the context clauses/sentences with which they occur has been used as a training signal for sentence representation learning. The authors used handcrafted rules for each marker in order to ensure that the markers signal an actual relation. DisSent has underwhelming results on the GLUE tasks as a STILT (Wang et al., 2019a).

**Discovery** (Sileo et al., 2019) is another dataset for discourse marker prediction, composed of 174 discourse markers with $10k$ usage examples for each marker. Sentence pairs were extracted from web data, and the markers come either from the PDTB or from a heuristic automatic extraction.

**DiscEval** refers to all DiscEval tasks used in a multitask setup; since we use a uniform loss weighting, we discard Persuasion classes other than Strength (note that the other classes can be considered subfactors for strength) in order to prevent the Persuasion task to overwhelm the others.

## 4.2 HUMAN ACCURACY ESTIMATES

For a more insightful comparison, we propose derivations of human accuracy estimates for the datasets we used.

The authors of SarcasmV2 (Oraby et al., 2016) dataset directly report $80\%$ annotator accuracy compared to the gold standard. Prasad et al. (2014) report $84\%$ annotator agreement for PDTB 2.0, which is a lower bound of accuracy. For GUM (Zeldes, 2017), an *attachment accuracy of* $87.22\%$ *and labelling accuracy of* $86.58\%$ *as compared to the 'gold standard' after instructor adjudication* is reported. We interleaved attachment and labelling in our task. Assuming human annotators never predict the non-attached relation, $69.3\%$ is a lower bound for human accuracy. Authors of the Verifiability (Park & Cardie, 2014) dataset report an agreement $\kappa = 0.73$ which yields an agreement of $87\%$ given the class distribution, which is a lower bound of human accuracy. We estimated human accuracy on EmoBank (Buechel & Hahn, 2017) with the intermediate datasets provided by the authors. For each target (V,A,D) we compute the average standard deviation, and compute the probability (under normality assumption) of each example rating of falling under the wrong category.

Unlike the GLUE benchmark (Nangia & Bowman, 2019), we do not yet provide human accuracy estimates obtained in a standardized way. The high number of classes would make that process rather more difficult. But our estimates are still useful even though they should be taken with a grain of salt.

## 4.3 OVERALL RESULTS

Task-wise results are presented in table 2. We report the average scores of 6 runs of STILT and finetuning phases.

DiscEval seem to be challenging even for the BERT base model, which has shown strong performance on GLUE (and vastly outperforms the baselines on our tasks). For many tasks, there is a STILT that significantly improves the accuracy of BERT. The gap between human accuracy and BERT is particularly high on implicit discourse relation prediction (PDTB and GUM). This task is known to be difficult, and previous work has shown that task dedicated models are not yet on par with human performance (Morey et al., 2017). Pretraining on MNLI does not improve the DiscEval average score for the BERT base model. A lower sarcasm detection score could indicate that BERT+MNLI is more focused on the literal content of statements, even though no STILT improves sarcasm detection. All models score below human accuracies, with the exception of emotion classification (but only for the valence classification subtask).

Table 3 shows aggregate results alongside comparisons with GLUE scores. The best overall unsupervised result (GLUE+DiscEval average) is achieved with Discovery STILT. Combining Discovery and MNLI yields both a high DiscEval and GLUE score, and also yields a high GLUE diagnostics score. All discourse based STILT improve GLUE score, while MNLI does not improve DiscEval average score. DiscEval tasks based on sentence pairs seem to account for the variance across STILTs.

|  | PDTB | STAC | GUM | Emergent | SwitchB. | MRDA | Persuasion | Sarcasm | Squinky | Verif. | EmoBank |
|---|---|---|---|---|---|---|---|---|---|---|---|
| CBoW | 27.4 | 32 | 20.5 | 59.7 | 3.8 | 0.7 | 70.6 | 61.1 | 75.5 | 74.0 | 64.0 |
| BiLSTM | 25.9 | 27.7 | 18.5 | 45.6 | 3.7 | 0.7 | 62.6 | 63.1 | 72.1 | 74.0 | 63.5 |
| BiLSTM+ELMo | 27.5 | 33.5 | 18.9 | 55.2 | 3.7 | 0.7 | 67.4 | 68.9 | 82.5 | 74.0 | 66.9 |
| Previous work | 48.2 | - | - | 73.1 | - | - | - | - | - | 81.1 | - |
| BERT | 48.8 | 48.2 | 40.9 | 79.2 | 38.8 | 22.3 | 74.8 | **77.1** | 87.5 | 86.7 | 76.2 |
| BERT+MNLI | 49.1 | 49.1 | 42.8 | 81.2 | 38.1 | 22.7 | 71.7 | 73.4 | 88.2 | 86.0 | 76.3 |
| BERT+DiscEval | 49.1 | **57.1** | 42.8 | 80.2 | **40.3** | **23.1** | **76.2** | 75.0 | 87.6 | 85.9 | 76.0 |
| BERT+DisSent | 49.4 | 49.0 | 43.9 | 79.8 | 39.2 | 22.0 | 74.7 | 74.9 | 87.5 | 85.9 | 76.2 |
| B+DisSent+MNLI | 49.6 | 49.2 | **44.2** | 80.9 | 39.8 | 22.1 | 74.0 | 74.1 | 87.6 | 85.6 | 76.4 |
| BERT+Discovery | 50.7 | 49.5 | 42.7 | **81.7** | 39.5 | 22.4 | 71.6 | 76.7 | 88.6 | 86.3 | **76.6** |
| B+Discovery+MNLI | **51.3** | 49.4 | 43.1 | 80.7 | **40.3** | 22.2 | 73.6 | 75.1 | **88.9** | **86.8** | 76.0 |
| Human estimate | **84.0** | - | **69.3** | - | - | - | - | **80.0** | - | **87.0** | 73.1 |

Table 2: Transfer test scores across DiscEval tasks; we report the average when the dataset has several classification tasks (as in Squinky, EmoBank and Persuasion); B(ERT)+$\mathcal{X}$ refers to BERT pretrained classification model after an auxiliary finetuning phase on task $\mathcal{X}$. All scores are accuracy scores except SwitchBoard/MRDA (which are macro-F1 scores). *Previous work* refer to the best scores from previous work that used a similar setup. PDTB score is from Bai & Zhao (2018). Emergent score is from Ferreira & Vlachos (2016). Verifiability score is derived from Park & Cardie (2014).

MNLI has been suggested as a good default auxiliary training task based on evaluation on GLUE (Phang et al., 2018) and SentEval (Conneau et al., 2017). However, our evaluation suggests that finetuning a model with MNLI alone has significant drawbacks.

More detailed results for datasets with several subtasks are shown in table 4. We note that MNLI STILT significantly decreases relevance estimation performance (on BERT base and while multi-tasking with DisSent). Many models surpass the human estimate at valence prediction, a well studied task, but interestingly this is not the case for Arousal and Dominance prediction.

|  | DiscEval$_{AVG}$ | D.E.-Pairs$_{AVG}$ | D.E.-Single$_{AVG}$ | GLUE$_{AVG}$ | GLUE$_{diagnostics}$ |
|---|---|---|---|---|---|
| BERT | 61.8±.4 | 57.9±.5 | 62.3±.3 | 74.7±.2 | 31.7±.3 |
| BERT+MNLI | 61.7±.5 | 57.2±.5 | 62.2±.4 | **77.0**±.2 | 32.5±.6 |
| BERT+DiscEval MTL | **63.0**±.4 | **60.0**±.4 | 62.6±.2 | 75.3±.2 | 31.6±.3 |
| BERT+DisSent | 62.0±.4 | 58.4±.4 | 62.2±.3 | 75.1±.2 | 31.5±.3 |
| B+DisSent+MNLI | 62.1±.4 | 58.2±.4 | 62.3±.2 | 76.6±.1 | 32.4±.0 |
| BERT+Discovery | 62.4±.3 | 58.2±.4 | 62.7±.3 | 75.0±.2 | 31.3±.2 |
| B+Discovery+MNLI | 62.5±.4 | 58.5±.5 | **62.8**±.3 | 76.6±.2 | **33.3**±.2 |

Table 3: Aggregated transfer test accuracies across DiscEval and comparison with GLUE validation downstream and diagnostic tasks (GLUE diagnostic tasks evaluate NLI performance under presence of linguistic phenomena such as negation, quantification, use of common sense); BERT+$\mathcal{X}$ refers to BERT pretrained classification model after auxiliary finetuning phase on task $\mathcal{X}$; D.E.-Pairs$_{AVG}$ is the average of DiscEval sentence pair classification tasks.

The categories of our benchmark tasks cover a broad range of discourse aspects. The overall accuracies only show a synthetic view of the tasks evaluated in DiscEval. Some datasets contain many subcategories that allow for a fine grained analysis through a wide array of classes (viz. 51 categories for MRDA). Table 5 shows a fine grained evaluation which yields some insights on the capabilities of BERT. We report the 5 most frequent classes per task. It is worth noting that the BERT models do not neglect rare classes. These detailed results reveal that BERT+MNLI scores for discourse relation prediction are inflated by good scores on predicting the absence of relation (possibly close to the neutral class in NLI), which is useful but not sufficient for discourse understanding. The STILTs have complementary strengths even with given tasks, which can explain why combining them is helpful. However, we used a rather simplistic multitask setup, and efficient combination of the tasks remains an open problem.

| | Persuasiveness | | | | EmoBank | | | Squinky | | |
|---|---|---|---|---|---|---|---|---|---|---|
| | Eloquence | Relevance | Specificity | Strength | Valence | Arousal | Dom. | Inf. | Implicature | Formality |
| BERT | 75.6 | 63.5 | 81.6 | 78.3 | 87.1 | 72.0 | 69.5 | 92.2 | 72.1 | 98.3 |
| BERT+MNLI | 74.7 | 57.5 | 82.3 | 72.2 | 86.6 | 72.4 | 69.9 | 92.5 | 73.9 | 98.1 |
| BERT+DiscEval | 75.6 | **64.0** | 83.2 | **82.0** | 86.8 | 71.9 | 69.2 | 92.3 | 71.8 | **98.6** |
| BERT+DisSent | 73.8 | 63.0 | 82.6 | 79.5 | 87.1 | 71.4 | 70.1 | 92.6 | 72.0 | 97.7 |
| B+DisSent+MNLI | 76.9 | 61.5 | **83.9** | 73.9 | **87.6** | 72.1 | 69.4 | 91.5 | 73.4 | 97.9 |
| BERT+Discovery | 76.0 | 59.1 | 80.1 | 71.4 | 86.8 | **72.6** | 70.5 | **93.2** | 74.2 | 98.5 |
| B+Discovery+MNLI | 74.1 | 60.4 | 79.4 | 80.4 | 86.4 | 72.1 | 69.6 | 93.1 | **75.3** | 98.4 |
| Human estimate | - | - | - | - | 74.9 | **73.8** | 70.5 | - | - | - |

Table 4: Transfer test accuracies across DiscEval subtasks (Persuasiveness, EmoBank, Squinky) BERT+$\mathcal{X}$ refers to BERT pretrained classification model after auxiliary finetuning phase on task $\mathcal{X}$.

| | BERT | B+MNLI | B+DisSent | B+Discovery | B+DiscEval | Support |
|---|---|---|---|---|---|---|
| GUM.no_relation | 48.9 | **51.0** | 46.0 | 45.4 | 43.3 | 48 |
| GUM.circumstance | 77.1 | **80.6** | 73.2 | 77.8 | 74.6 | 35 |
| GUM.elaboration | 41.5 | 38.5 | 40.0 | **46.1** | 42.9 | 32 |
| GUM.background | 22.6 | 25.3 | 34.3 | **38.2** | 35.8 | 23 |
| GUM.evaluation | 20.4 | 22.6 | **36.8** | 29.9 | 35.1 | 20 |
| STAC.no_relation | 59.9 | **63.8** | 55.4 | 61.3 | 46.9 | 117 |
| STAC.Comment | 77.8 | 76.1 | 74.9 | **78.6** | 54.4 | 115 |
| STAC.Question_answer_pair | 79.1 | 80.1 | **83.3** | 76.9 | 83.0 | 93 |
| STAC.Q_Elab | 32.1 | 34.3 | 32.0 | 38.1 | **63.7** | 86 |
| STAC.Contrast | 29.6 | 37.4 | 25.9 | 27.5 | **49.9** | 53 |
| SwitchBoard.Uninterpretable | 86.0 | 86.0 | 85.5 | 86.1 | **86.3** | 382 |
| SwitchBoard.Statement-non-opinion | 72.0 | 72.1 | **72.4** | **72.4** | **72.4** | 304 |
| SwitchBoard.Yes-No-Question | **85.9** | 85.2 | 85.5 | **85.9** | 85.8 | 303 |
| SwitchBoard.Statement-opinion | 46.3 | 46.3 | 48.6 | 48.8 | **49.5** | 113 |
| SwitchBoard.Appreciation | **73.5** | 71.1 | 70.2 | 71.7 | 72.9 | 108 |
| PDTB.Cause | 55.2 | 55.7 | 53.1 | **57.2** | 55.9 | 302 |
| PDTB.Restatement | 40.4 | 40.0 | 41.3 | **43.9** | 41.0 | 263 |
| PDTB.Conjunction | 52.8 | **53.9** | 52.1 | 53.3 | 52.5 | 262 |
| PDTB.Contrast | 45.8 | **49.0** | 47.2 | 48.0 | 46.0 | 172 |
| PDTB.Instantiation | 56.6 | 55.6 | 52.8 | **58.7** | 55.7 | 109 |
| MRDA.Statement | 51.2 | 51.8 | 48.9 | **53.4** | 51.4 | 364 |
| MRDA.Defending/Explanation | 52.8 | 54.1 | **55.3** | 52.8 | 52.0 | 166 |
| MRDA.Expansions of y/n Answers | **51.7** | 48.7 | 50.3 | 49.6 | 49.4 | 139 |
| MRDA.Offer | 48.6 | 46.9 | **50.7** | 49.4 | 49.4 | 102 |
| MRDA.Rising Tone | 39.3 | 40.1 | 40.3 | **40.7** | 38.8 | 98 |

Table 5: Transfer F1 scores across the categories of DiscEval tasks; B(ERT)+$\mathcal{X}$ denotes BERT pretrained classification model after auxiliary finetuning phase on task $\mathcal{X}$.

## 5 CONCLUSION

We proposed DiscEval, a set of discourse related evaluation tasks, and used them to evaluate BERT finetuned on various auxiliary finetuning tasks. The results lead us to rethink the efficiency of mainly using NLI as an auxiliary training task. DiscEval can be used for training or evaluation in general NLU or discourse related work. Much effort has been devoted to NLI for training and evaluation for general purpose sentence understanding, but we just scratched the surface of the use of discourse oriented tasks. In further investigations, we plan to use more general tasks than classification on sentence or pairs, such as longer and possibly structured sequences. Several of the datasets we used (MRDA, SwitchBoard, GUM, STAC) already contain such higher level structures. In addition, a more inclusive comparison with human annotators on discourse tasks could also help to pinpoint the weaknesses of current models dealing with discourse phenomena. Yet another step would be to study the correlations between performance metrics in deployed NLU systems and scores of the automated

evaluation benchmarks (GLUE/DiscEval) in order to validate our claims about the centrality of discourse.

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

APPENDIX A

| dataset | $N_{dev}$ | $N_{test}$ | $N_{labels}$ | labels |
|---|---|---|---|---|
| PDTB | 1204 | 1085 | 16 | [Cause, Conjunction, Restatement, Contrast,Instantiation,Asynchronous ...] |
| GUM | 259 | 248 | 17 | [Elaboration, No-Relation, Circumstance, Condition,Concession, ...] |
| STAC | 1247 | 1304 | 18 | [No-Relation, Question-Answer-Pair, Comment,Acknowledgement, ...] |
| MRDA | 1630 | 6459 | 50 | [Statement, Defending/Explanation, Expansions Of Y/N Answers, ...] |
| SwitchBoard | 2113 | 649 | 41 | [Uninterpretable, Statement-Non-Opinion, Yes-No-Question, ...] |
| Emergent | 259 | 259 | 3 | [For, Observing, Against] |
| Sarcasm | 469 | 469 | 2 | [Not-Sarcasm,Sarcasm] |
| Verifiability | 634 | 2424 | 3 | [Unverifiable, Non-Experiential, Experiential] |
| Persuasiveness-Specificity | 62 | 62 | 2 | [Low, High] |
| Persuasiveness-Eloquence | 91 | 90 | 2 | [Low, High] |
| Persuasiveness-Relevance | 91 | 90 | 2 | [Low, High] |
| Persuasiveness-Strength | 46 | 46 | 2 | [Low, High] |
| Squinky-Formality | 453 | 452 | 2 | [Low, High] |
| Squinky-Informativeness | 465 | 464 | 2 | [Low, High] |
| Squinky-Implicature | 465 | 465 | 2 | [Low, High] |
| EmoBank-Arousal | 684 | 683 | 2 | [Low, High] |
| EmoBank-Dominance | 798 | 798 | 2 | [Low, High] |
| EmoBank-Valence | 644 | 643 | 2 | [Low, High] |

Table 6: Number of labels ($N_{labels}$), development examples ($N_{dev}$), test examples ($N_{test}$) of DiscEval tasks)

