# OpenReview forum: "Discourse-Based Evaluation of Language Understanding"
_ICLR.cc/2020/Conference — Reject_

### Official Review · AnonReviewer3 · 2019-10-20
**Official Blind Review #3**

**Rating:** 6

**Review:**


The paper presents a new GLUE-like dataset collection called DiscEval, which focuses on discourse and pragmatics. Like GLUE and SuperGLUE, datasets from existing works are collated and formated. The tasks are all classification tasks.

The paper also evaluates several baselines including bag-of-words, BiLSTM encodings, and BERT fine-tuned on different types of data, which have shown success in GLUE tasks.

Like other NLP benchmarks, the DiscEval benchmark would be a good resource for other researchers to hill-climb their systems on, provided that the data format is standardized and the submission system is easy to use like GLUE.

That said, the paper has some rooms for improvement:

- With the information in Table 2, it is hard to judge the difficulty and headroom for each task. Only a few tasks have human evaluation scores were estimated from the inter-annotator agreement. Contrast this to the GLUE and SuperGLUE papers which provide human baselines from actual humans. Without these anchoring numbers, it is hard to see if the remaining gap is due to the model's inability to model discourse, or due to noise in the dataset.

- Providing the best single-task result from previous work would also help give a more complete picture.

- With the result of fine-tuned BERT almost matching the human performance in several tasks, the argument that BERT is not a universal representation (abstract + introduction) is weakened somewhat.

As a valuable resource for other researchers, I am still leaning toward acceptance despite the issues above.

Other comments:

- The bullet points on Page 5 could be clarified. Currently, the first bullet seems to contain multiple groups, for example.

- Sentiment analysis (in GLUE) could be viewed as a discourse task. It would be nice to be a bit more upfront about it.

**Experience Assessment:**

I have read many papers in this area.

**Review Assessment: Checking Correctness Of Derivations And Theory:**

I assessed the sensibility of the derivations and theory.

**Review Assessment: Checking Correctness Of Experiments:**

I assessed the sensibility of the experiments.

**Review Assessment: Thoroughness In Paper Reading:**

I read the paper at least twice and used my best judgement in assessing the paper.

---

> ### Author Response · Authors · 2019-11-14
> **Response to Review #3**
>
> Many thanks for your helpful comments and constructive criticism.
>
> - With the information in Table 2, it is hard to judge the difficulty and headroom for each task. Only a few tasks have human evaluation scores were estimated from the inter-annotator agreement. Contrast this to the GLUE and SuperGLUE papers which provide human baselines from actual humans. Without these anchoring numbers, it is hard to see if the remaining gap is due to the model's inability to model discourse, or due to noise in the dataset.
>
> > See general answer regarding human annotations and headroom.
>
> - Providing the best single-task result from previous work would also help give a more complete picture.
> > We added a “previous work” row into table 2, but not all previous work used a comparable setup.

---

### Official Review · AnonReviewer2 · 2019-10-23
**Official Blind Review #2**

**Rating:** 6

**Review:**

- Overview: This work presents a benchmark for language understanding centered on understanding pragmatics and discourse, in contrast to existing NLP benchmarks which focus on mostly semantic understanding. The stated contributions are
    - The DiscEval benchmark, with some estimates of human performance
    - Baseline performance with state of the art models
    - Comparisons between commonly used supervised training objectives (NLI) and discourse-oriented objectives

- Review: This work offers a complementary evaluation benchmark for NLU systems to what currently exists in the field, and also offers compelling evidence that the current methods used do not provide good signal for learning the types of phenomena assessed in this benchmark. I recommend (weak) accept.
    - The paper would benefit a lot from having more in-depth explanation of the tasks, what the classes are, and what the classes mean. It's not always clear from the examples or the labels in Table 1 what exactly is being tested.
        - Additionally, it'd be nice to get the source of the data for each task.
    - The rough grouping of tasks in nice.
    - I understand that estimates of human performance, especially for tasks with a high number of classes, can be tricky to obtain. I do feel that they are especially important to have for this dataset. As you said, some of these tasks rely on context that isn't explicitly provided in the utterance, so I wonder if for some of the tasks, there may be insufficient context in just the provided utterances for humans to perform well.
        - Additionally, some tasks seem to have quite subjective label definitions. For example, "eloquence" and "specificity" in Persuasion; "formality" and "informativeness" in Squinky; etc. Validating that humans are consistent on these tasks seems crucial.
        - For the estimates you do have, are these expert or lay annotators? For some of these annotations, it seems like you'd need trained annotators.
    - Have you given any thought to exploitable data artifacts that may occur in these datasets that might be driving the fairly high performance on some of these tasks (e.g. Verifiability)?
    - I think it's quite interesting that fine-tuning on MNLI doesn't lead to good performance on DiscEval, as MNLI, as the authors point out, is commonly taken to be a useful pretraining task. This discrepancy gives practical weight to the authors' claim that discourse and pragmatic phenomena are not being sufficiently studied or evaluated for in current NLP research, despite the fact that these handling these phenomena will be crucial for NLP systems.

- Things to Improve + Questions + Comments
    - I would hope that most people in the NLP community would not say that language understanding is a solved problem, but I agree that putting "universal" in model names and paper titles is a reach-y thing to do.
    - Tasks
        - What is the original citation for STAC?
    - S4.3, Table 2:
        - Any reason not to try combining all four pretraining/intermediate training tasks in a single model, or at least more combinations of DiscEval with other things?
        - Could you comment on the standard deviation of the scores (per task) given that you're averaging 6 runs?
    - Table 3
        - "The best overall unsupervised result is achieved with Discovery STILT": what does unsupervised mean here? It also doesn't look like BERT+Discovery is the best in any column here.
    - Tables 4 and 5 feel like a bit of a dump of numbers. It'd be more useful to the readers to extract trends (probably using the groupings and theoretical frameworks introduced earlier). The noting of BERT+MNLI being good at predicting absence of relations is nice.
    - Typos: There are a noticeable number of typos. Here are some I noted:
        - The formatting of the task descriptions in Section 3 is a bit inconsistent and awkward.
        - Table 1: "exemple"
        - P4: "We use a custom train/dev validation split"
        - P5: "that help *realize* speech acts' intentions."
        - P6: "Prediction of discourse markers based *off* the context clauses..."


**Experience Assessment:**

I have published one or two papers in this area.

**Review Assessment: Checking Correctness Of Derivations And Theory:**

N/A

**Review Assessment: Checking Correctness Of Experiments:**

I carefully checked the experiments.

**Review Assessment: Thoroughness In Paper Reading:**

I read the paper at least twice and used my best judgement in assessing the paper.

---

> ### Author Response · Authors · 2019-11-14
> **Response to Review #2**
>
> Many thanks for your helpful comments and constructive criticism.
>
> - Additionally, some tasks seem to have quite subjective label definitions. For example, "eloquence" and "specificity" in Persuasion; "formality" and "informativeness" in Squinky; etc. Validating that humans are consistent on these tasks seems crucial.
> > Some labels are indeed more subjective than others. The annotations of the tasks you cite were derived from multiple annotations, which was one of our criteria for selecting datasets. The degrees of agreement are imbalanced but always above chance. Squinky and Persuasion factors were initially rated with continuous ratings. The fact that we discretized them and removed the middle quantile should further alleviate the subjectivity problem.
>
> - Have you given any thought to exploitable data artifacts that may occur in these datasets that might be driving the fairly high performance on some of these tasks (e.g. Verifiability)?
> > The CBoW and LSTM baselines should be able to use respectively lexical information and simple structural patterns (e.g. sequence length). The fact that BERT outperforms them with such a margin shows that it is using more than that. The fact that fine-tuning with DiscEval and with Discovery improves Disceval results shows that there is a common ground between some (but not all) tasks, which indicates either other common artifacts or actual linguistic capabilities, but further work would be needed to really address this problem.
>
> - Any reason not to try combining all four pretraining/intermediate training tasks in a single model, or at least more combinations of DiscEval with other things?
> > Combining DiscEval with datasets like MNLI is possible, but could be tricky because MNLI/Discovery/Dissent are quite large, and our simplistic multi-task setup would either not use the full datasets or overfit the small datasets. Such combinations would be interesting but would involve more complicated setups.
>
> - Could you comment on the standard deviation of the scores (per task) given that you're averaging 6 runs?
> > Most stds are below 1% absolute, so the 6 runs averaging should allow meaningful comparisons
>
>  - "The best overall unsupervised result is achieved with Discovery STILT": what does unsupervised mean here? It also doesn't look like BERT+Discovery is the best in any column here.
> > The Discovery and Dissent datasets have been obtained without human annotations, hence the term unsupervised; we meant that it was the best according to an average of GLUE and DiscEval. We clarified this in the paper.

---

> > ### Comment · AnonReviewer2 · 2019-11-14
> > **Response to Response to Review #2**
> >
> > - "Additionally, some tasks seem to have quite subjective label definitions."
> > >> I see. The points you make seem reasonable, and I agree that there are probably significant technical challenges in obtaining human annotations that might go beyond the scope of this work (but I do hope you keep working on them! I think it would help adoption as researchers like having a goal performance to reach).
> >
> > - "Have you given any thought to exploitable data artifacts that may occur in these datasets that might be driving the fairly high performance on some of these tasks (e.g. Verifiability)?"
> > >> The CBoW and LSTM baselines are a good start, but I was thinking more along the lines of partial input biases (e.g. strong performance of hypothesis-only baselines on MNLI). It isn't crucial, but would be nice to check out (maybe added to an appendix) in light of a surprising number of recent datasets and benchmark with these biases.
> >
> > - "Any reason not to try combining all four pretraining/intermediate training tasks in a single model, or at least more combinations of DiscEval with other things?"
> > >> I see. In the past when I've had a similar issue with varying dataset sizes, I've gotten away with sampling tasks proportional to the amount of training data, which avoids overfitting on really small datasets and seeing enough of the large datasets. By the magic of neural networks and multi-task training, the model learns on all datasets without the large datasets overwhelming the small ones. Might be worth a shot in the future.
> >
> > - "Could you comment on the standard deviation of the scores (per task) given that you're averaging 6 runs?"
> > >> Gotcha. I would still push for their inclusion (again, maybe in an appendix) since some of the results seem within 1% absolute (MRDA, Squinky, Verif, EmoBank), especially if you have them already.

---

### Official Review · AnonReviewer1 · 2019-11-02
**Official Blind Review #1**

**Rating:** 6

**Review:**

This paper presents a new benchmark for natural language understanding called DiscEval. The benchmark focuses on datasets that more directly measure a model's understanding of the discourse structure and relations in the text. The paper makes direct comparisons to GLUE and especially studies the previous claims of MNLI being a generically good pre-training task. They find that MNLI pretraining, using STILTS, does not help BERT's performance on DiscEval.

Some comments and recommendations,
- You say that for PDTB you "select the level 2 relations as categories," I believe these are the class level relations. Maybe add in a brief explanation in the paper?
- Was discarding all but the Strength subclass from the Persuasion dataset and empirically motivated decision or just something you did a-priori?
- The human results are already hedged, but maybe the grain of salt should be bigger: it occurs to me that the comparison of the model performance and human performance could be unfair since the human performance is reported on the original tasks, before the filtering of data and changing label distribution like with PDTB and GUM.
- Particularly given the data filtering and restructuring of some of the tasks, getting a rough estimate of human performance for all the datasets would be quite valuable
- I think saying the MNLI does not help model performance on DiscEval is completely valid but claiming it hurts performance could be a stretch given the margins of error.
- I know tasks like PDTB and GUM have quite a few classes. Please include the number of classes for each of the datasets in the benchmark. Also including information on the size of the dev and test sets could be helpful.

Overall, I agree with the author's claim that we need to have a broader evaluation suite for NLU, and this benchmark is a step in the right direction. Some of the DiscEval datasets measure NLU information that is somewhat orthogonal to what datasets in GLUE/SuperGLUE measure. I think this will be a useful resource to the research community


Minor things,
- Page 5, last line the CoLA citation is wrong, should be Warstadt et al. (2019)

**Experience Assessment:**

I have published one or two papers in this area.

**Review Assessment: Checking Correctness Of Derivations And Theory:**

N/A

**Review Assessment: Checking Correctness Of Experiments:**

I assessed the sensibility of the experiments.

**Review Assessment: Thoroughness In Paper Reading:**

I read the paper at least twice and used my best judgement in assessing the paper.

---

> ### Author Response · Authors · 2019-11-14
> **Response to Review #1**
>
> Many thanks for your helpful comments and constructive criticism.
>
> - You say that for PDTB you "select the level 2 relations as categories," I believe these are the class level relations. Maybe add in a brief explanation in the paper?
>
> > Level 2 corresponds to type level relations. It would have been possible to use the finer grained (level 3) relations but they are not available for all types, forcing us to either discard examples or have heterogeneous labels. Level 2 is also the main current target of dedicated discourse models.
>
> - Was discarding all but the Strength subclass from the Persuasion dataset and empirically motivated decision or just something you did a-priori?
> > We discard classes other than Strength only when using DiscEval as a multi-task training signal. It was an a priori decision motivated by the use of a uniform weighting per task. Classes other than persuasion were kept during evaluation, but using them all for training could have been redundant, and would give too much weight to this task.
>
> - The human results are already hedged, but maybe the grain of salt should be bigger: it occurs to me that the comparison of the model performance and human performance could be unfair since the human performance is reported on the original tasks, before the filtering of data and changing label distribution like with PDTB and GUM.
> > We did not affect class distribution in the PDTB and took a worst case scenario for the estimate of GUM human accuracy.

---

### Author Response · Authors · 2019-11-14
**General Response to Reviewers**

We thank the reviewers for their detailed and constructive feedback; we will provide answers to common concerns below, and respond to specific concerns in separate replies to each reviewer. We updated the paper in order to take into account all suggestions.

Obtaining human estimates is feasible when the number of labels is limited. As can be seen in table 6, dialog act classification tasks have many labels, whereas the estimates proposed after the GLUE publication [Nangia19] often deal with 3 labels. With many labels, the name and descriptions of labels could have more weight during annotations, which makes the situation different from  [Nangia19].

The question of headroom has also been raised; the human estimates show that there is headroom even though they could be too optimistic. But the fact that specific finetuning (especially DiscEval as a multi-task training signal) improves DiscEval scores shows that there is headroom at least for NLU models that are not fine-tuned.

[Nangia19] Human vs. Muppet: A Conservative Estimate of Human Performance on the GLUE Benchmark
Nikita Nangia, Samuel R. Bowman
Proceedings of the 57th Annual Meeting of the Association for Computational Linguistics

---

### Decision · Program_Chairs · 2019-12-19

**Decision:**

Reject

**Comment:**

This paper proposes a new benchmark to evaluate natural language processing models on discourse-related tasks based on existing datasets that are not available in other benchmarks (SentEval/GLUE/SuperGLUE). The authors also provide a set of baselines based on BERT, ELMo, and others; and estimates of human performance for some tasks.

I think this has the potential to be a valuable resource to the research community, but I am not sure that it is the best fit for a conference such as ICLR. R3 also raises a valid concern regarding the performance of fine-tuned BERT that are comparable to human estimates on half of the tasks (3 out of 5), which slightly weakens the main motivation of having this new benchmark.

My main suggestion to the authors is to have a very solid motivation for the new benchmark, including the reason of inclusion for each of the tasks. I believe that this is important to encourage the community to adopt it. For something like this, it would be nice (although not necessary) to have a clean website for submission as well. I believe that someone who proposes a new benchmark needs to do as best as they can to make it easy for other people to use it.

Due to the above issues and space constraint, I recommend to reject the paper.